

# Mechanism of generation and propagation characteristics of coastal trapped waves in the Black Sea

Müjdat Aydın and Şükrü Turan Beşiktepe

Institute of Marine Sciences and Technology, Dokuz Eylul University, 35340, Izmir, Turkey

*Correspondence to*: Şükrü Turan Beşiktepe (sukru.besiktepe@deu.edu.tr)

**Abstract.** Coastal trapped waves (CTW) were investigated in the Black Sea during 2012–2016 using observations from the sea level stations at five locations along the southern coast of the Black Sea together with the sea surface height model from
Copernicus Marine Service. Spectral and wavelet analysis of sea level data shows that CTWs exist in the Black Sea with a period of 12–13 days and 15 days duration. These waves propagate from west to east with a speed of 2.3-2.6 m s$^{-1}$ and cause 10-20 cm variability in sea level. To investigate formation mechanisms of CTWs, sea surface height and surface velocities from Copernicus Marine Service, wind measurements from sea level stations and atmospheric model results from Copernicus Marine Service are jointly analyzed. These analyses showed that CTWs were formed when water accumulated on the western
shelf after gale force alongshore winds blowing in the western Black Sea. Our results provide clear observational evidence on process of the excitement of CTWs by wind stress. CTWs generate a coastal current reaching up to 0.5 m s$^{-1}$. This coastal current joins to large-scale cyclonic inertial current flowing over the continental slope and accelerates it. Hence, we present evidence on the influence of the CTWs on the large scale circulation.

## 20  1 Introduction

The existence of trapped modes in certain cases was first discovered by Stokes **(**1846**),** who obtained trapped wave solution on fundamental mode edge waves on a sloping beach; these waves propagate in the longshore direction and their amplitude decays exponentially in the offshore direction. This theory was extended a century later by Ursell (1951) to include the whole spectrum of possible modes. These initial studies were mainly concerned with edge waves without rotation whose typical
frequencies are above the inertial frequency. Reid (1958) studied the effect of the rotation on shallow water edge waves and showed that unidirectionally propagating waves with the coast to the right (left) in the northern (southern) hemisphere at sub-inertial frequencies arise.

Coastal trapped waves were first observed by Shoji (1961) as the coastal sea level disturbances of several days period moving southward along the east Japanese coast and low frequency long waves propagating clockwise along the Australian coast by
Hamon (1962). Hamon analyzed mean sea level and atmospheric pressure fluctuations at some stations along the east Australian coast and noticed that at very low frequencies the daily sea level on the shelf did not respond as an inverse barometer





as would be expected. Hanon found that the sea level was depressed only about half the expected response. He also found that the spectra of the adjusted sea level were peaked at six and nine days, corresponding to peaks in the atmospheric pressure spectrum. The results suggested the presence of a low frequency nondispersive left bounded wave which travelled along the continental shelf. Robinson (1964) developed a model based on the linearized shallow water equations with rotation and variable topography to explain the phenomenon observed by Hamon (1962). Robinson (1964) has suggested that low frequency waves are generated as a response of sea level to atmospheric pressure on the Australian coasts. Robinson called these waves as Continental Shelf Waves and shelf wave theory was proposed. Adams and Buchwald (1969) have shown that wind stress is the driving mechanism for the low frequency response of the sea surface rather than the pressure changes due to the large-scale moving weather system. Later, it was shown that the alongshore component of the wind stress was a dominant factor for generation of continental shelf waves (Gill and Schumann, 1974). The model has been extended with offshore stratification, variable bottom, and longshore current by Mysak (1967) and Gill and Clarke (1974). It is shown that stratification and topography with coastal boundary support the existence of both shelf waves and internal Kelvin waves. The free coastal modes are a hybrid between the two and Gill and Clarke (1974) proposed to call them as coastal trapped waves. Coastal trapped waves were investigated in depth after these initial theoretical and experimental developments. An extensive review of these earlier studies and developments can be found in Mysak (1980). Following these pioneering studies, further theoretical and observational efforts have been rapidly pursued leading to the deeper characterization of coastal trapped waves (Brink, 1991; Huthnance, 1995).

It is now well known that the margins of the ocean act as an efficient waveguide for the propagation of CTWs from the region of their excitation. Typically mode-1 CTWs have maximum amplitude on the shore and decays exponentially offshore with the scale of the Rossby radius of deformation. They can freely propagate very long distances from the formation area with the coast on the right (left) in the Northern (Southern) Hemisphere, with periods ranging from a few days to weeks, without changing their character. Hence, CTWs are a major mechanism to distribute the energy from the atmosphere in the ocean. It was shown theoretically and observationally that Coastal trapped waves (CTW) play a significant role in large scale, low frequency sea level, and current variability on the continental shelf and slope areas (Huthnance, 1995). Although CTWs are produced by different mechanisms, those in Kelvin mode are formed by winds blowing parallel to the coast by accumulating water to the shore through Ekman transport (Adams and Buchwald, 1969; Gill and Schumann, 1974). Observational evidence of CTWs forced by the longshore wind has been documented all around the margins of the ocean since it was first observed in the 1960s. Examples of observations of CTWs at sub-inertial frequencies induced by storms include along the west coast of South America (Zamudio, 2002; Romea and Smith, 1982), the west coast of North America (Beckenbach and Washburn, 2004), the coast of South Africa (Schumann and Brink, 1990), along the Japanese coasts (Kitade, 2000; Igeta et al., 2007), around Australia and New Zealand (Maiwa et al., 2010; Stanton, 1990), along the west coast of India (Amol et al., 2012), in the East China Sea (Yin et al., 2014) and so on. It has been shown by these observations that CTWs have typically 8-16 days period with 2-4 m s$^{-1}$ phase speeds and have O (10 cm) amplitudes on the coast.





Despite the significant role of low frequency CTWs in the coastal dynamics, observational studies on the coastal trapped waves in the Black Sea are limited, being confined to the northern coast (Fig. 1). Besides, these studies were based on short-duration measurements carried out during spring-summer periods (Ivanov et al., 2015). To our knowledge, the generation mechanism of mode-1 Coastal Trapped Waves (CTWs) propagating along the coast of the Black Sea and their roles on the dynamics of the large scale circulation have not been studied.

In their study based on analysis of current measurements made on the Crimean shelf, Ivanov and Yankovsky (1993) found that 11-12 day oscillations in coastal currents have the largest amplitude, leading to a 15–20 cm s$^{-1}$ increase in the alongshore component of the velocity. As a result of the analysis, it was postulated that these oscillations were produced by distant winds on a spatial scale comparable to the length of the Black Sea and were trapped on the shore with a phase velocity of 2 ms$^{-1}$. Ivanov and Bagaiev (2014) used a 3-d regional model in this area to investigate a wide range of oscillations in sea level and

temperature. Ivanov and Bagaiev (2014) noted that oscillations at 120, 260, and 360 h periods in the sea level at the coast have statistically significant spectral energy. They explained these oscillations as Kelvin waves or a response of the shelf water to synoptic winds. CTWs propagating westward in this area scatter at the southernmost tip of the Crimea Peninsula due to the coastline and topographic variations and anticyclonic eddies could be developed downstream (Yankovsky and Chapman, 1995, 1997).

In basin-scale modelling studies on the Black Sea current system, the presence of coastal trapped waves and their possible effects on the current was also stated. Rachev and Stanev (1997) performed numerical experiments using the primitive equation model, which found that general cyclonic circulation in the Black Sea can be formed even in conditions of weak cyclonic wind vorticity, due to the propagation of coastal trapped waves. Staneva et al. (2001) detected eastward-propagating CTWs along the southern boundary from model results.

The Black Sea comprises an elliptic shaped deep basin curved on the major axis. Its major axis extends 1180 km in the east-west direction and the minor axis extends 264 km north-south. The shelf on the western part of the sea constitutes approximately 20 per cent of the whole sea. The width of the shelf gradually narrows toward the southwestern corner of the basin and terminates to the east at 31 E. on the southern boundary. The coastal regions of the rest of the basin have a narrow continental shelf (approximately 20 km) connected to a deep abyss with a steep slope.

The large scale circulation of the Black Sea is cyclonic with a strong inertial current over the continental slope around the basin. The well-defined Black Sea cyclonic boundary current (rim current) flows over the continental slope with a mean velocity of 30 cm s$^{-1}$ (Oğuz and Beşiktepe, 1995). Classically, cyclonic wind patterns (positive wind stress curl) and the inflows of freshwater that originate from the large rivers on the Northwest part of the Black Sea are postulated as the main forces for cyclonic surface circulation. The presence of the rim current flowing cyclonically is expected to be modified by

long waves propagating along the same direction. On the other hand, CTWs are expected to play an important role in the stability of the black sea rim current and the formation of mesoscale variability.





The objective of this study is firstly to provide observational evidence for CTWs in the Black Sea using a series of sea level data along the southern coast and then to identify their generation mechanism and demonstrate their impact on the rim current.

The paper is structured as follows: observed data sets and numerical model results used in this study are described in section
2. Section 3 introduces the identification of CTW from sea level records along the Turkish coast. In section 4, generation mechanisms of the observed CTWs are identified, and in Section 5 impacts of the CTWs on the coastal current will be evaluated using model results.

## 2 Data

In situ sea level data were obtained from the Turkish National Sea Level Monitoring System (TUDES) operated by the Turkish General Directorate of Mapping along the Black Sea coast of Turkey. There are five stations (İğneada, Şile, Amasra, Sinop, and Trabzon) along the Black Sea coast of Turkey from west to east, respectively (Fig. 1). The data were collected at 15 min. intervals at local datum. The raw data is processed to remove outliers and then cleaned data are binned into hourly sea levels.

Sea level and surface currents from the Black Sea Reanalysis of Physical Fields (BS-Currents) from Copernicus Marine
Environment Monitoring Services (CMEMS, http://marine.copernicus.eu) are used to reveal the spatial extent of the observed CTWs and the role of the CTWs in the Black Sea rim current. The model used in CMEMS is based on the Nucleus for European Modelling of the Ocean (NEMO, v3.4) with horizontal grid resolution 1/36° zonally, 1/27° meridionally (ca. 3 km) and 31 unevenly spaced vertical levels. The observations assimilated in the BS-Currents using variational assimilation include in situ profiles, along-track sea level anomalies (SLA) and gridded sea surface temperature (SST) provided by Copernicus
TACs.

Hourly wind data is obtained from the Turkish State Meteorological Service in proximity to sea level stations. Wind fields are also provided by the Copernicus Marine Environment Monitoring Services. They are estimated from ASCAT and OSCAT scatterometers retrievals and from ECMWF operational wind analysis with a spatial resolution of 0.25 ° and 6 h in time, and available at synoptic time 00h:00; 06h:00; 12h:00; 18h:00. (Bentamy and Fillon, 2012)


## 3 Identification of CTWs in the Black Sea

Because amplitude of mode-1 CTWs is greatest on the coast, these waves can be inferred from coastal sea level measurements. The sea level obtained between January 2012 and January 2017 from the stations of the TUDES network in the Black Sea is shown in Fig. 2 after the mean and trend are removed. Sea levels at all stations synchronously vary on different time scales
ranging from from a few days (meteorological scale) to seasonal and annual due to different physical processes acting on different timescales. Changes in sea level in the Black Sea show an obvious seasonal cycle; the highest sea level occurs between spring and summer, while the lowest is seen in fall. This seasonal cycle is in accordance with the seasonal change in freshwater





entering the Black Sea. Increasing river inflows increase the sea level of the Black Sea in the spring and the lowest sea level is in autumn when river flows are the minimum. The outflow from the Black Sea is controlled by the flow through the Turkish

Straits, and changes in river influxes could be felt in sea level. Moreover, interannual variability of sea levels is also evident and can be attributed to the decadal changes in freshwater influxes from rivers. In addition to these variations in long time scales, energetic variations at sea level, which occur at shorter than monthly timescales, are visible. These energetic events, which are of interest in this study, can be attributed to atmospheric forcing.

To detect the dominant frequencies of variability in the sea level time series to qualitatively examine the changes in sea level,

the variance-preserving spectra of sea level are calculated at all stations (Fig. 3). Spectral analysis using the Welch method was conducted on the hourly binned sea level data.

Although tidal amplitudes are small in the Black Sea, diurnal and semidiurnal (not shown in this figure) tidal frequencies were found to be evident in sea level spectra at all stations. As we move to lower frequencies, i.e., longer periods, clear peaks in the low frequency region (0.07 – 0.16 cpd or 5-14 day period) of the spectrum are visible. A sharp peak in low frequency spectra

occurs at a occurs at 14.2 days (0.07 cpd), followed by 5-6 days (0.15 to 0.19 cpd). The peaks in the spectra at periods of 5-15 days correspond to weather band, indicating atmospheric forcing. These values are in good agreement with the numerical calculations of wave properties in the Black Sea (Ivanov and Bagaiev, 2014; Ivanov et al., 2015).

The spectral analysis results given above reveal that periods of 5-15 days were predominant in the sea level time series at all stations in the southern Black Sea. The spectral analysis assumes that processes are stationary in time and hence does not give

information on the nonstationary parts of the signal. However, due to the seasonal variation in atmospheric forcing, nonstationarity in sea level changes should be expected, particularly in the weather band. Wavelet analysis expands time series into time-frequency space and then is convenient to identify the time dependent signal characteristics of sea level oscillations. In this study, the Continuous Wavelet Transform (CWT) is based on the Morlet wavelet function applied to the sea level using the wavelet toolbox (MATLAB) developed by (Grinsted et al., 2004). The results are presented in Fig. 4.

Power is seen to reach a maximum at low frequencies with 10 to 15 day periods between the autumn-winter months at all sites. These periods are well matched with the results obtained from the spectral analysis presented above. The years 2015 to 2016 in Sinop and 2012 in Amasra are exceptions, due to missing data (see Fig. 2). However, this portion of the missing data does not prevent us from seeing the overall structure.

One of these low frequency sea level variations was formed during October-November 2014 period and we selected this period

as a case to perform a detailed analysis to reveal characteristics of low frequency waves and their impact on the Black Sea circulation.

Fig. 5 shows variations of the sea level with time during one month between 15 October and November 15, 2014. At all stations, the sea level first evidently decreased and then attained a maximum of 20 cm. Characteristically, the sea level reaches the peak level in 2 days after the sea level is minimum. This 2-day lag is the same for all stations. The wave crest observed in





Şile on October 26, 2014, emerged in Amasra on October 28, in Sinop on October 30, and in Trabzon on November 1. This visual inspection of the time series of sea level shows that the sea level fluctuation propagates eastward, and the sea level signal is not modified during this propagation. There are clear time lags between the sea level responses at each station. In this case, we can say that a wave from west to east has progressed and reached Trabzon from Şile in about 5-6 days.

Lagged cross correlation of sea level between the first station on the west (Sile) and stations towards the east demonstrate the
propagating nature of the observed oscillations (Fig. 6). Maximum lagged correlations between the westernmost station (Şile) and the other stations towards the east (Amasra, Sinop, Trabzon) occurs at days 1.4, 2.5 and 5.2, respectively. Distances Şile-Amasra, Şile-Sinop and Şile-Trabzon are 280 km, 510 km, and 1010 km, respectively.  By using time delayed correlations and inter-station distances, the phase velocity of the wave is calculated as approximately 2.3 m s$^{-1}$ and does not change much during its inter-station journey. Our calculations for periods other than October-November 2014 (not shown here) gave phase velocity
of sub-inertial CTWs in the Black Sea is in the range of 2–3 m s$^{-1}$. These values indicative of mode 1 CTW (Hughes et al., 2019) and are comparable with the phase velocities of 11-12 day oscillations in the Black Sea estimated by Ivanov and Yankovsky (1993).

## 4 Mechanism of Generation of the Observed CTWs in the Black Sea

To understand the mechanisms of generation of the observed CTWs as described above, wind data coinciding with the same period were examined. The temporal variation of hourly wind at the westernmost station (İğneada), from 15 October to 15 November 2014 is shown in Fig. 7.

Wind speed varied periodically on 3–4 day time scales, changing direction 180 degrees after each relaxation. The region is mainly dominated by the recurrence of the north wind blowing, alternating with weak wind periods. The frequent change in
direction indicates the passage of fronts over the area. A strong wind with a speed of more than 8 m s$^{-1}$ often occurred and wind speed attained its maximum (>18 m s$^{-1}$) on October 25 from the north-eastern direction. Spatial distribution of this storm for the period of October 24–27, 2014 presented in Fig. 7 spans the duration of south winds that change direction to strong north-easterly and reach 18 m s$^{-1}$.  This gale force wind parallel to the coastline from the northeast is favourable to downwelling and piles up the water to the shore. Before and after this gale force wind from the northeast, two wind patterns are visible; before,
the wind over the area fluctuated few times from downwelling favourable (north-easterly) to upwelling favourable (south-easterly), each lasting approx. 2 days. After the storm, the wind changes direction to north-westerly. This peak in wind speed corresponds to the peak in the sea level at İğneada.

The wind stress, as resulting from the meteorological model is presented in Fig. 8. The wind field obtained from CMEMS is in good agreement with the corresponding time series recorded at the coastal station. It should be noted that the coastal station
is located at the southern end of the core of the storms.





The winds are rather weak, directed from E-S in the whole Black Sea basin on October 24, 2014. On 25 October 2014, the wind suddenly changed direction and increased intensity, blowing at gale force from the northeast in the western part of the basin. The intensity of the wind increased on 26 October, showing an intensity maximum in the north-western and central parts of the Black Sea. Toward October 27, 2014, the intensity of the wind gradually decreased while keeping its direction. Note that the wind direction was oriented along the coastline during the storm.

Fig. 9 shows the evolution of the sea surface height (SSH) during the event presented above. The strongest storm occurred on October 25, and the sea level responded quickly. On 25 October, surface waters start to accumulate along the western boundary of the Black Sea with strong north-easterly winds, resulting in Ekman transport toward the coast. On the following day, this layer becomes wider and extends toward the southwestern boundary. Joint analysis of the winds (ref. Fig. 7 and Fig. 8) and sea level data showed that the accumulation of water at the coast begins when the wind speed exceeds 8 m s$^{-1}$.

The event was preceded by moderate wind conditions. The coastal sea level due to Ekman transport reached an anomaly of 0.3 m. The accumulated waters propagated consistently south through October 27 after the winds weakened and changed direction toward the NW.

## 5 Impact of CTWs on the Black Sea general circulation

Figure 10 shows CMEMS horizontal distributions of the current vector at the surface coinciding with Figs. 8 and 9. On October 24, the surface current in the Black Sea is weak except in some parts of the northern boundary. On 25 October, surface currents on the western Black Sea started to increase and south-westward velocities of about 0.5 m s$^{-1}$ occurred near the coast of the western boundary associated with a 0.3 m rise in the mean water level. The accumulated water at the coast, coinciding with the wind change direction on October 25, creates a pressure gradient directed offshore, and longshore currents are generated toward the south. On 26 October, this current reached the southern boundary. This strong current progressively moved along the southern boundary of the basin reaching central parts on October 27. On November 1, the strong flow was visible all the way to the south-east corner of the basin. The maximum alongshore velocity reached up to 1 m s$^{-1}$ and the cross shore scale is about 40 km. After Amasra, the shelf width is narrower, and the cross shelf scale of the current stretches about 20 km.

## 6 Summary and Concluding Remarks

Sea level measurements from five coastal stations situated in the southern Black Sea are analyzed to reveal low frequency oscillations in the basin. The sea level oscillations have a 20 cm range in the subinertial frequency band with 12-13 day periodicity. Generation of coastal trapped waves along the western coast of the Black Sea and their propagation along the southern boundary are demonstrated using CMEMS models.

During the October-March period, eastward-travelling depressions produce gale force winds from the north (Özsoy and Ünlüata, 1997). These gale force winds trigger a persistent downwelling along the western boundary of the Black Sea and the accumulation of water at the coast. Following the relaxation of the wind, coastal trapped waves at sub-inertial frequencies propagate eastward along the southern boundary, keeping the coast on the right. These waves are produced in the western part

of the basin during the winter when the wind speeds exceed 12 m s$^{-1}$. This phenomenon was occurring a few times a year.

The internal Rossby radius of deformation (20-30 km) is larger than the width of the continental shelf (20-25 km) along the southern and eastern boundary of the Black Sea. When the shelf scale is comparable with the internal Rossby radius, as, in the southern and eastern Black Sea, the margins of the Black Sea act as a vertical wall and become an efficient waveguide for the propagation of internal Kelvin waves. The waves formed have maximum amplitude on the shore and decay exponentially

offshore with the scale of the Rossby radius of deformation.

The coastal trapped waves produce current up to 0.5 m s$^{-1}$ in magnitude along the western and southern boundary of the Black Sea. Black Sea rim current flowing over the continental slope come closer to the coast because of the narrowing shelf along the southern boundary. Hence, the transient strong currents generated by CTWs interact with this rim current. Since both are cyclonic, the rim current is intensified. This suggests that the intensification of the Black Sea mean circulation during winter

is associated with the coastal trapped waves generated by the alongshore winds on the western boundary. Observations of the CTWs on the northern boundary of the Black Sea reported in Ivanov and Yankovsky (1993) were possibly generated on the western boundary of the Black Sea.

**Data availability**

In situ sea level data used in this study can be obtained from https://tudes.harita.gov.tr. The CMEMS data can be obtained from https://marine.copernicus.eu/.

**Author contribution**

MA conducted the research as a part of his master thesis supervised by STB. STB wrote the paper with input from MA.

**Competing interests**

The authors declare that they have no conflict of interest



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



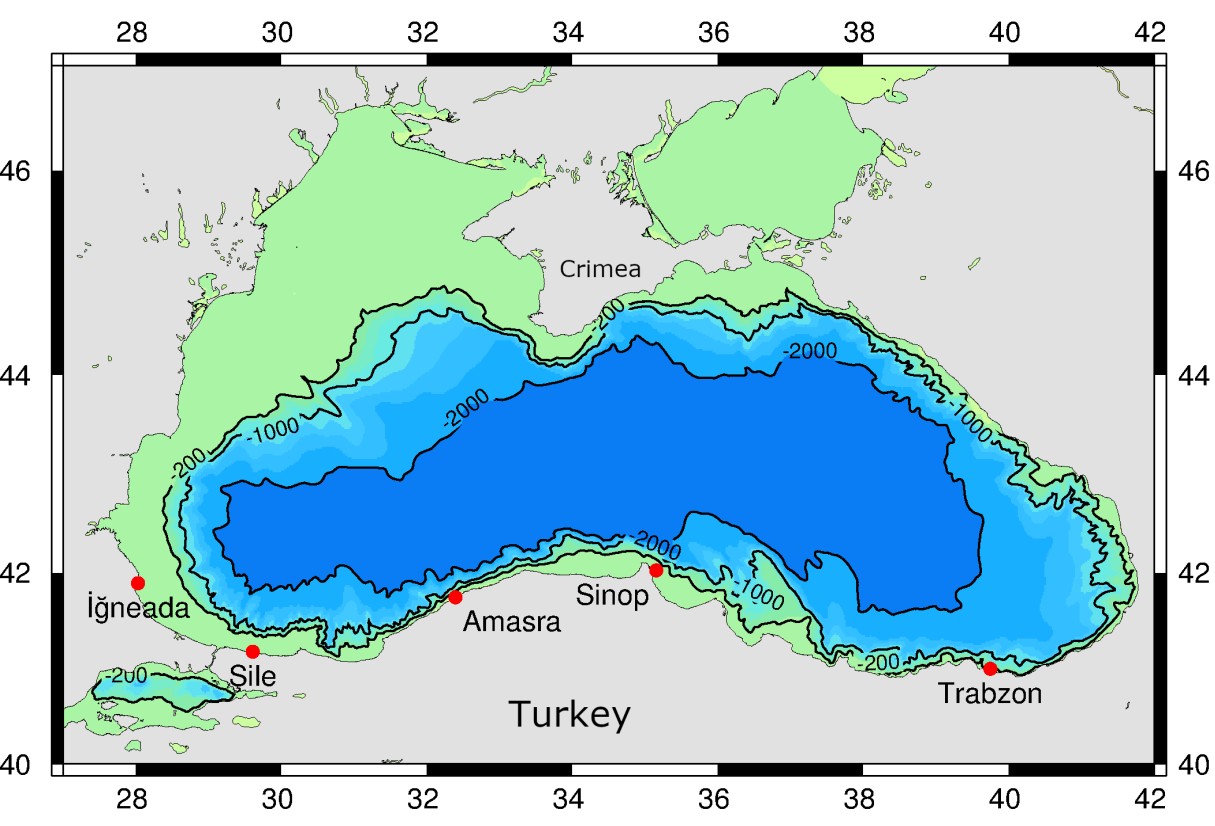

Figure 1: Locations of tide gauge stations and bathymetry (m) of the Black Sea.





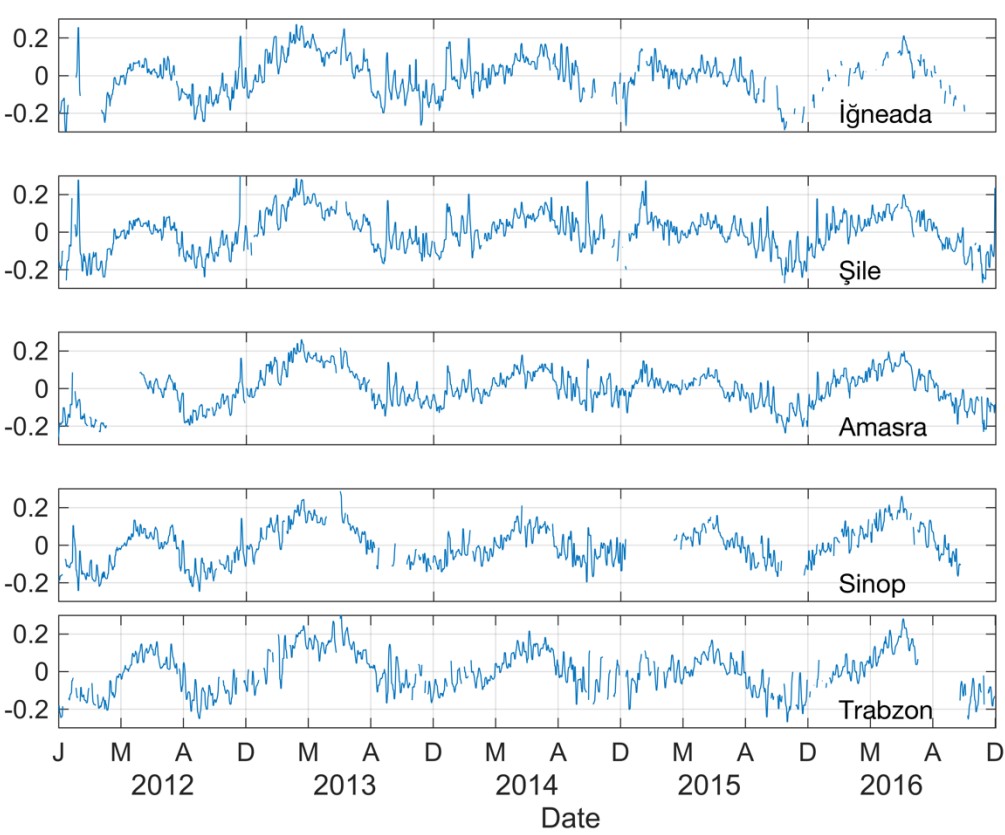


**Figure 2. Time series of hourly sea level from İğneada, Şile, Amasra, Sinop and Trabzon from 2012 to 2017.**







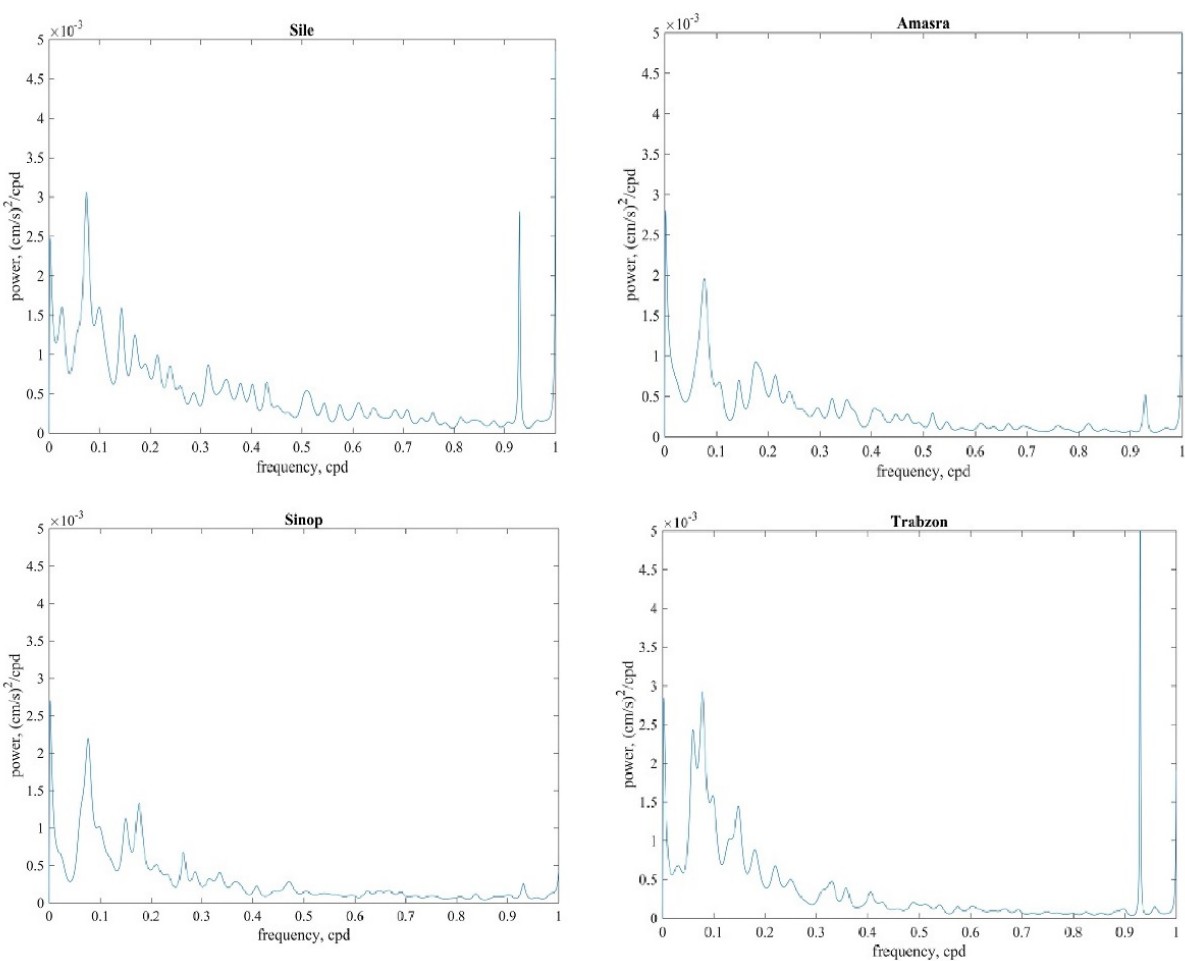

**Figure 3. Variance preserving spectra of the sea level for İğneada, Şile, Amasra, Sinop and Trabzon; frequencies in cycles per day (cpd).**





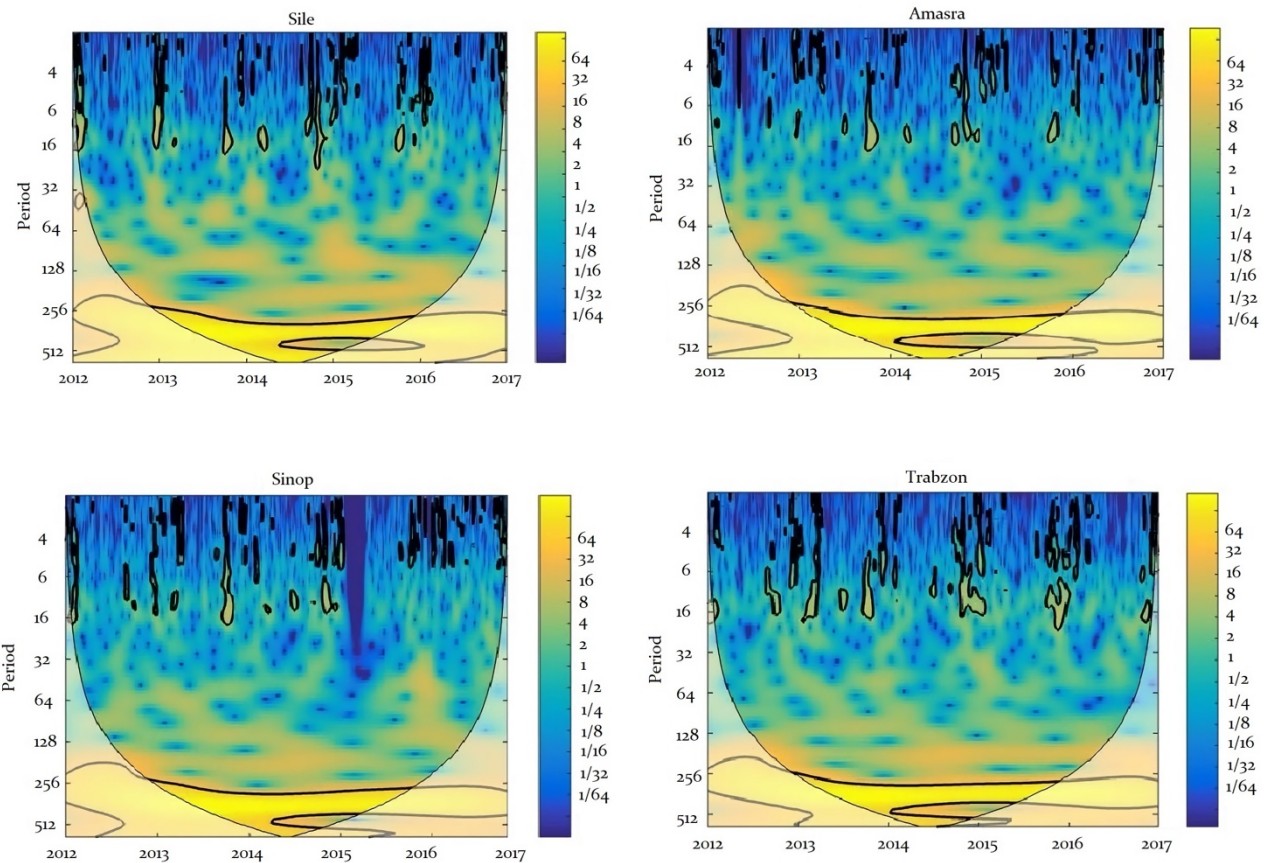

**Figure 4. Wavelet power spectrum of the sea level for Şile, Amasra, Sinop and Trabzon, using the Morlet wavelet. Time is indicated on the x-axis (years) and the timescale (period) on the y-axis. The colour scale of the variance (power) on the z-axis represents an increasing power (variance) from blue to yellow. The areas enclosed by the black contour lines indicate the periods with significance above 95%.**








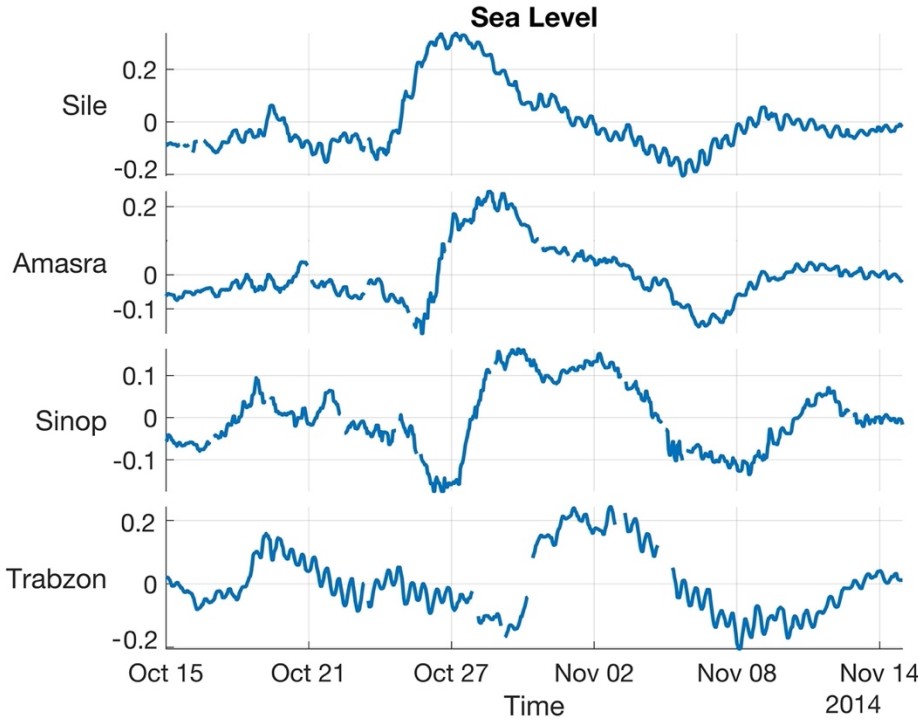

**Figure 5.** Time series of the sea level variations from October 15, 2014 to November 15, 2014. The data is subsampled from Fig. 2.





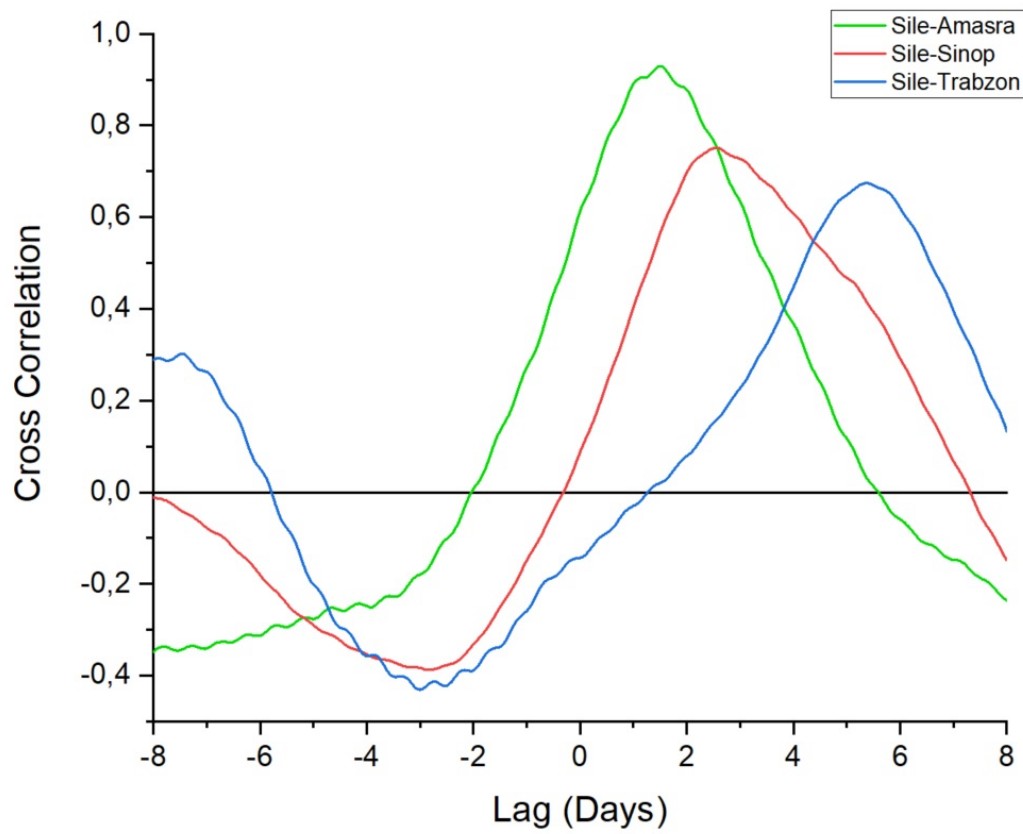

**Figure 6. Time-lagged correlations between sea level at westernmost station (Sile) and other stations towards east for October-November 2014.**

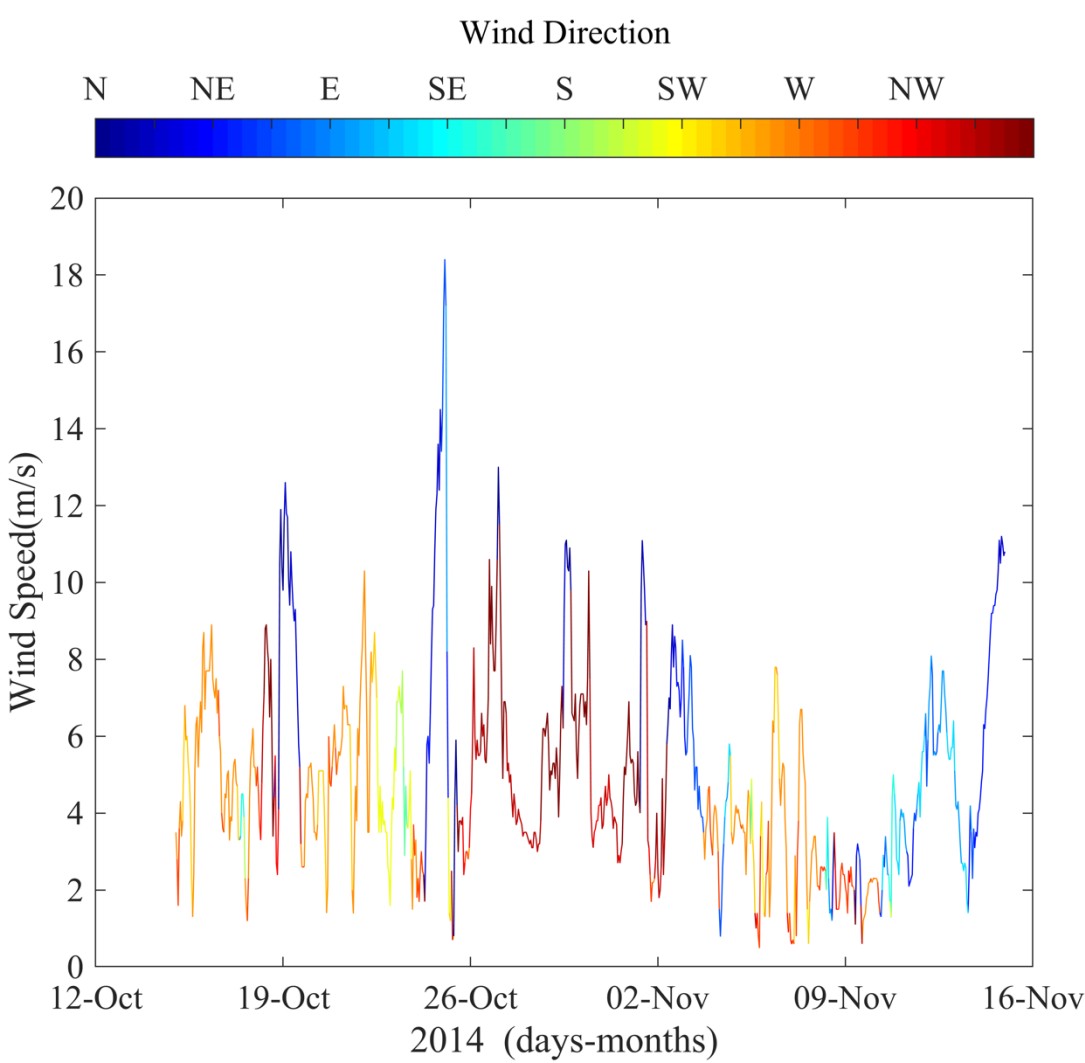

**Figure 7. Wind speed and direction at İğneada for the October 2014 event.**



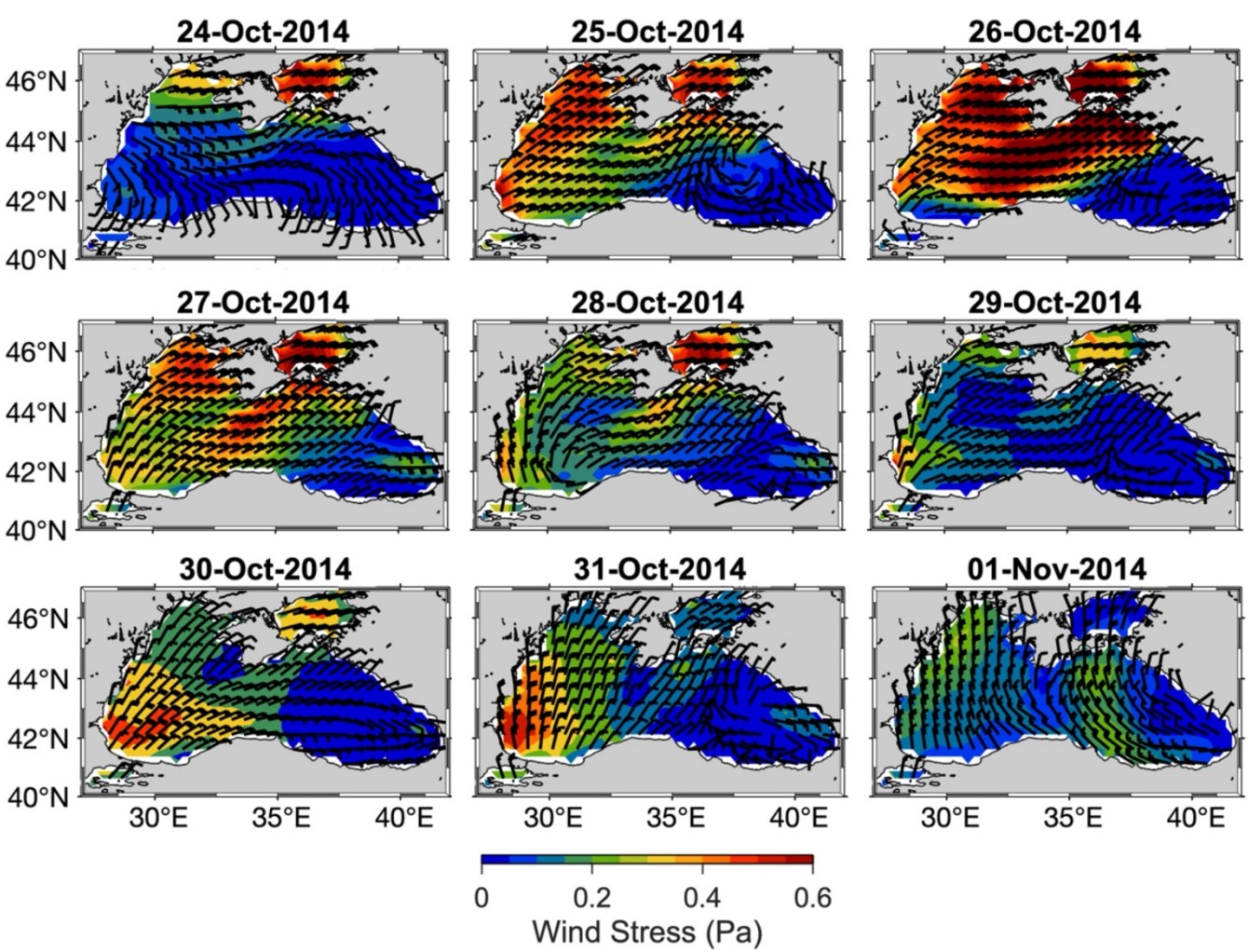

**Figure 8. Sequence of wind stress (colour scale) and direction (wind barbs) distribution from 24 October to 01 November 2014 at 00:00 UTC .**



**Figure 9. Sequence of daily mean sea surface height 24 October to 01 November 2014 obtained from Black Sea reanalysis product of CMEMS.**



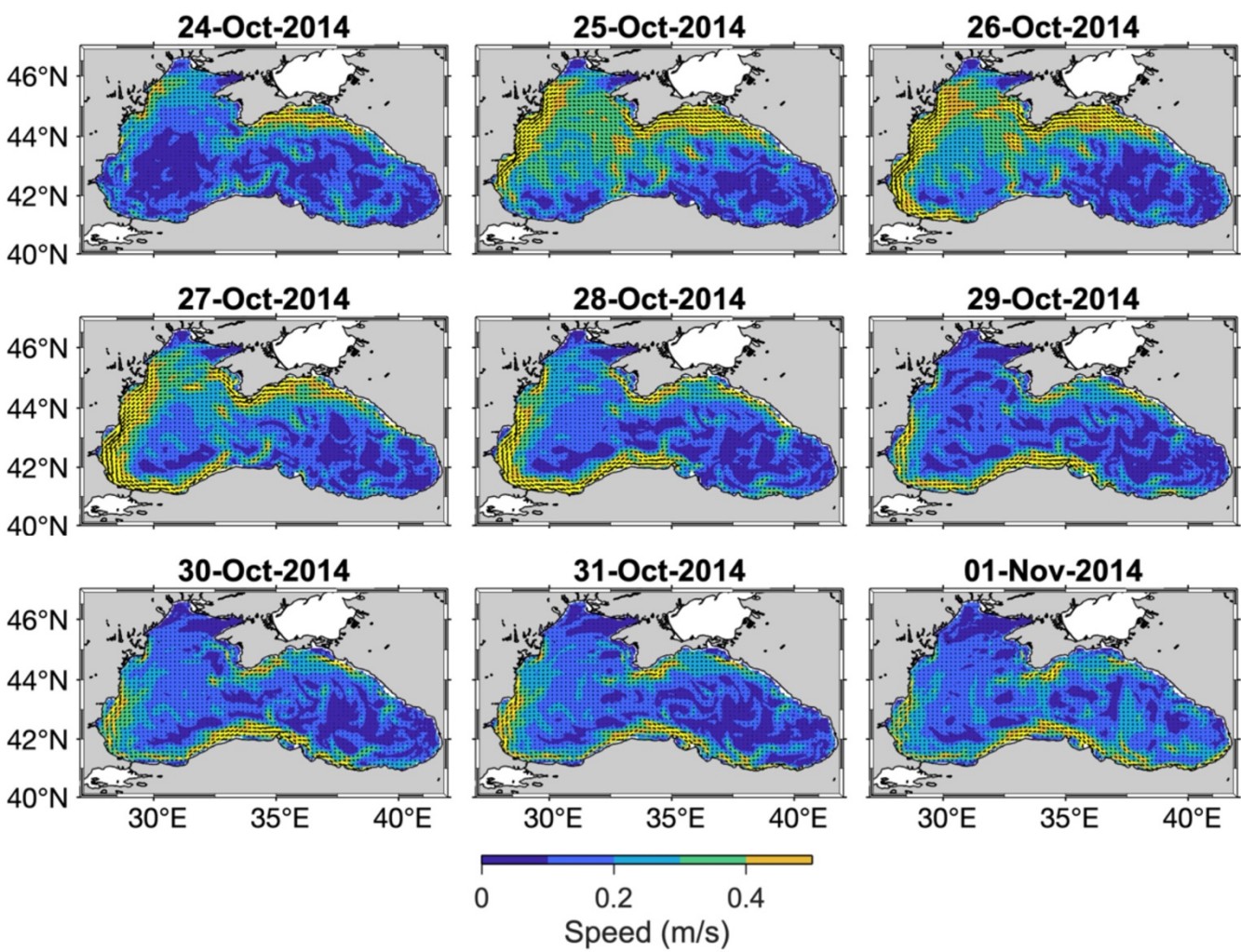

**Figure 10. Sequence of daily mean horizontal velocity distributions at 2.5 m from 24 October to 01 November 2014 obtained from Black Sea reanalysis product of CMEMS.**