# Peer review of "Mechanism of generation and propagation characteristics of coastal trapped waves in the Black Sea"

_Ocean Science, 2022_

## Author Response (AR2)

Dear Editor,

Thank you very much for your effort to edit our manuscript. We did all corrections in the final revised form.

For Aydın (2019), we gave link from another repository, we hope you will be able to accessthis time.

Discrepancy in the text due to the Fig 7- Fig8 corrected. Fig 7 improved.

Last para in the summary and conc. section modified and moved to end of section 5 to make it more clear.

We would like to thank to Reviewers for their helpful comments that led to clarifications and improvements in the manuscript. Please find our specific response (in italic) to the Reviewers comments below each reviewer's statements.

**Anonymous Referee #1**

- In the abstract, please state the knowledge gap more clearly and highlight the novel findings of your study. This becomes more clear in the introduction, but could/should be emphasized in the abstract.

*Abstract re-written according to your suggestion*

- L9: Consider referring to this data set as "reanalysis" instead of "model" throughout the manuscript.

*Model replaced with reanalysis*

- The introduction is very long relative to the length of the entire manuscript. Consider tailoring it more to the background of your study (i.e., shorten the general part in the beginning and/or move some of the context from the Black Sea to a more extended discussion section).

*Historical development of the subject removed from introduction and some infor about Black Sea moved to Section 5 and conclusion.*

- L75: Please state the periods of the oscillations in days, not hours, to be consistent with the rest of the manuscript and to make it easier for the reader to compare the values.

*Corrected*

- L83f: How do the results of Staneva et al. (2001) compare to your study? This would be interesting to elaborate on in the discussion section.

*We added new references for comparision with numerical modelling studies.*

- L90: I am wondering about the "inertial current" - is the rim current not in geostrophic balance?

*It is in geostrophic balance. Inertial remover to avoid confusion.*

- L95f: This is something that could be addressed in more detail in Section 5. In particular, be explicit about how the stability may be affected (weakening vs. strengthening). Are there any topographic features where the waves could scatter (like the Crimea Peninsula in Yankovsky and Chapman, 1955, 1997)? Are there any local "hot- spots" of eddy kinetic energy (which could be inferred from, e.g., satellite altimetry)?

*We removed this sentence from the text to avoid confusion.*

- L116: Please provide an URL or DOI for the hourly wind data in the data availability section.

*We added URL to the data availability section.*

- L121: Maybe "southern Black Sea" is more appropriate.

*This suggestion taken and "southern" is added*

- L131: Can you provide a reference for that the changes in freshwater influxes from rivers are decadal (versus interannual variability)? This is not obvious.

*Decadal is written by mistake. It replaced with nonseasonal and the reference provided (Volkov and Landerer, 2015)*

- L133: Please provide a reference if there is a study that has shown the link to the atmospheric forcing (or reword and refer to your section 4).

*Sentence is reworded with referencing to section 4.*

- You show time series from five stations in Fig. 2, but focus on only four stations afterwards. Please state explicitly in the text why you do not include further analysis from AᵒẌ̈ neada.

*Results of the iğneada added to Figures 3 and 4. This was unintentional. However, İğneada is not shown in Figure 5 due to missing data.*

- The title and abstract indicate general results, but most findings are based on a case study for one of the CTW events in October-November 2014. The provided evidence of the case study is not sufficient to make general conclusions. Why did you not analyze all events you identified using the wavelet analysis? This would strengthen the manuscript. The use of composite figures could help to visualize results. If you choose to not extend your analysis, be more clear in that your conclusions mostly are based on a case study - this should be reflected in both the title and abstract.

*We added reference Aydın(2019) documenting 3 cases which are very similar. To avoid repeating at each figure, we present here one case. This reference is available on the internet and easily accessible.*

- L205ff: This section would benefit from a discussion of context from existing literature. In the introduction, you mention some relevant papers. Some of that information could be moved here and compared to your results.

*We extended the section using existing literature.*

- L222-225: These are general results for which no robust direct evidence is provided in the manuscript. In L225 you mention that this occurs more often, but this is not shown or mentioned where you present the results.

*This was not our result and refence was given in the beginning of the paragraph.*

- L229f: The exponential decay is something we theoretically expect (here it sounds like a statement from the literature). You can connect this to your results by discussing this explicitly in Section 5 - the strongest velocities and highest sea level anomaly are found near the coast according to Figs. 9-10.

*This is taken into consideration while revising section 5.*

- L234: This is interesting, but you have not mentioned the intensification of the Black Sea mean circulation during winter before. Are there any references or observations to substantiate this statement?

*References from model and observations added*

**Technical corrections**

- Fig. 1: Please add a schematic arrow of the Black Sea cyclonic boundary current to the map and add labels (°N/°E) to the axes.

This figure re-drawn according to your suggestion.

- Fig. 2: The time series look rather smooth for being in hourly resolution. If the data were filtered, please mention this in the caption and methods section. Please add labels including units to the y-axes.

*Description of the smoothing are given now both in figure caption and in the text.*

- Fig. 3: The caption states that these are spectra from all five stations, but only four panels are shown. Could you indicate some level of statistical significance? Consider also adding a second x-axis on top of each panel showing the period in days, as this is what you mostly refer to in the text. It would also be helpful if you could mark the main periods, for example by using a background shading.

*Fifth panel added and periods marked*

- Fig. 4: Please specify that the unit for Period is days.

*done*

- Fig. 5: Please add labels including units to the y-axes (or title).

*Labels added*

- Fig. 6: On the y-axis, please use period not comma to indicate decimals as in the rest of the manuscript.

*Corrected*

- Fig. 7: I strongly recommend to avoid use of a rainbow colormap like Jet. See e.g., Crameri et al. (2020), https://www.nature.com/articles/s41467-020-19160-7.pdf. Please also add a space between speed and (m/s). You could consider shading the background for the period 24-27 October, which you discuss in more detail in the text.

*Thank you for this info. We replaced the colormap and the period 24-27 October shaded*

- Figs. 8-10: Please indicate the locations of Ａ°Ａ¨ neada and Amasra, which you mention directly in the text when discussing these maps. (You could maybe even indicate all five stations as in Fig. 1.)

*Figures are already very crowded and instead of adding names we refer to coordinates in the text.*

- Fig. 9: Please clarify the caption ("mean sea surface height" vs. "SLA" in the color bar label).

*Caption corrected.*

- Fig. 10: You could consider plotting fewer (larger) arrows to enhance legibility.

*Number of arrows decreased by half and we made them bigger.*

*All the corrections on the language done.*

**Referee #2 Emil Stanev**

1. In case that authors keep in the revised manuscript the first paragraph (~line 25), this is perhaps the place where they can also mention Kelvin waves.

*Introduction shortened with removing historical develepments (first paragraph).*

2. You can consider mentioning in paragraph, lines 70-80, that basin wide numerical experiments aimed to studying coastal shelf waves have been carried out by Stanev and Beckers (1999) and use part of what they found as support to what you study. Even better is that you explain what exactly step ahead you do in comparison with the old studies. Paragraph, line 140, shows that similar periods were found by these authors too.

*Thank you very much again for reminding this publication. We added this reference in introduction and benefit from its results in result and conclutions sections.*

3. Add units in Fig. 4 and its caption. Check for the same problem all figures, for instance fig. 5 etc.

*Units added*

4. Explain in more detail how to read and understand Fig. 4.

*Detailed explanation of the Fig. 4 is given in the caption. We initially repeated this explanation in the text, but topic editor found it unnecessary*

5. Mark in Fig. 2 the time period presented in Fig. 5. Actually, Fig. 5 is the most important figure in this manuscript. Along with Figure 3, it deserves deeper analysis. More fundamental is to ask whether there is only one clear event or coastal wave propagation during the period presented in Fig. 2.

   *We added reference Aydın(2019) documenting 3 cases which are identical. To avoid repeating at each figure, we present here one case. This reference is available on the interned and easily accessible.*

6. The expression "Spatial distribution of this storm" (~line 180) is unclear from the graph in Fig. 7. May be Fig. 8? Explain how the graph illustrates spatial distribution.

   Thanks. This was a mistake and corrected in the revised version.

7. One basic problem is that the analysis of model results is not coherent with the analysis of observations. I wonder whether authors can find similar propagation characteristics as in figure 5, but sampled from the model data. This would be better than showing figure 10. What does spectral analysis of model data show?

*As we replied earlier;*

*Your recommendation on the deeper analysis with model results seems reasonable. However, we could not do it even we also thought before;*

*i. The model outputs from CMEMS was available one snapshot per day and hence the time resolution was not enough.*

*ii. We did not want to go into the deeper analysis of the model which we do not have full control over it. This can be done in collaboration with CMEMS model developers as follow-up work. We wanted to focus on observations and demonstrate the existence of the phenomena in the Black Sea and its role in the Black Sea dynamics in a descriptive manner.*

8. Some statements cannot be derived from the analysis: Line 230 "The waves formed have maximum amplitude on the shore and decay exponentially offshore with the scale of the Rossby radius of deformation." . You can check that using model results.

   *You are right. We corrected the text and indicating that this is from literature.*

9. Although English is not my mother tongue, I find that the text needs substantial improvement by native English speaking scientist.

*I beleive, corrections by other referee and editör improved the English of the text.*

---

## Author Response (AR3)

Dear Editor,

Thank you very much for your letter. We made all the minor corrections suggested by referees. Comments of the referee 1 on the weathear band and Kelvin wave are confusing and we prefer do not elaborate further. We added the word "synoptic" in Line 112 which might it clear.

We also added funding and acknowledge sections.

Yours sincerely,

---

## Author Response (AR4)

Dear Editor,

Thank you very much for clarifying comments of the referee with rewording. We beleive our terminology created confusion and we tried to clarify it.  We removed the "The peaks in the spectra at periods of 5-15 days correspond to the weather band, indicating synoptic atmospheric forcing" sentence and added a specific conclusions of the previous model studies.  Because it was giving a wrong impression as relating 14 days oscillations to atmospheric oscillation.

In order to be consistent with the results section, we also replaced "internal Kelvin waves" with "coastal trapped waves in mode-1" in the conclusion.

Sorry for creating confusion and we hope that these changes will make our presentation more clear.

Yours sincerely.

---

## Author Response (AR5)

Dear Editor,

We are grateful for your patience and help throughout the review process. We did technical corrections in the final form of the manuscript.

Yours sincerely,